# The effect of travoprost on primary human limbal epithelial cells and the siRNA-based aniridia limbal epithelial cell model, *in vitro*

Shuailin Li[1,2]*, Tanja Stachon[1,2], Shanhe Liu[1,2], Zhen Li[1], Shao-Lun Hsu[1,2], Swarnali Kundu[1], Fabian N. Fries[1,3], Berthold Seitz[3], Maryam Amini[1], Shweta Suiwal[1,2], Nóra Szentmáry[1,2]

**1** Dr. Rolf M. Schwiete Center for Limbal Stem Cell and Congenital Aniridia Research, Saarland University, Homburg/Saar, Germany, **2** Department of Experimental Ophthalmology, Saarland University, Homburg/Saar, Germany, **3** Department of Ophthalmology, Saarland University Medical Center, Homburg/Saar, Germany

* shuailinli97@gmail.com

## Abstract

### Purpose

Haploinsufficiency of the PAX6 gene is the primary pathogenic mechanism underlying classical congenital aniridia. Notably, at least 50% of patients with this condition develop glaucoma. Prostaglandin analogues, such as travoprost, are widely used to lower intraocular pressure in this patient population. At the same time, limbal epithelial cells (LECs) are believed to play a key role in the development and progression of aniridia-associated keratopathy (AAK). Therefore, the aim of this study was to investigate the effects of travoprost on cell viability, proliferation, migration, and the expression of key genes and proteins in both normal and PAX6-knockdown LECs.

### Materials and Methods

Primary human LECs were isolated from seven corneal donors. To simulate the haploinsufficient state characteristic of congenital aniridia, a PAX6-knockdown model, using siRNA was employed. LECs were treated with 0.039–40 µg/mL travoprost concentration for 20 minutes. Cell viability was assessed using the XTT assay, while cell proliferation was evaluated by the BrdU assay. Based on XTT results, 0.156 and 0.313 µg/mL travoprost were selected for further measurements in both LECs and PAX6-knockdown LECs. A scratch assay was conducted to measure cell migration. The expression levels of PAX6, FOSL2, MAPKs, inflammatory markers, caspase-3, and MMP9 were analyzed at both the gene and protein levels using qRT-PCR, Western blot, and ELISA.

**Data availability statement:** All relevant data are within the manuscript and its Supporting information files.

**Funding:** The author(s) received no specific funding for this work.

**Competing interests:** The authors have declared that no competing interests exist.

## Results

Travoprost significantly reduced LEC viability at 0.156 µg/mL (p = 0.028), while only higher concentrations (20 and 40 µg/mL) inhibited significantly LEC proliferation (p ≤ 0.004). PAX6-knockdown LECs exhibited reduced migration compared to control cells (p ≤ 0.046); however, treatment with 0.313 µg/mL travoprost significantly enhanced their migration (p = 0.047), accompanied by upregulation of JNK1/2 protein (p = 0.039) and MMP9 mRNA and protein levels (p = 0.021, p = 0.027). PAX6 knockdown led to suppression of inflammation-related genes (p ≤ 0.031) and travoprost did not exacerbate inflammatory responses (p ≥ 0.155). Additionally, 0.313 µg/mL travoprost significantly increased caspase-3 protein levels in PAX6-knockdown LECs (p = 0.044).

## Conclusions

Travoprost, at specific concentrations, can reduce the viability and proliferation of limbal epithelial cells. At 0.313 µg/mL, it significantly upregulates JNK1/2 and MMP9 expression, thereby enhancing the migratory capacity of PAX6-knockdown LECs. These findings may offer valuable insights for the selection of antiglaucomatous medications in patients with congenital aniridia.

## 1 Introduction

Congenital aniridia is a rare hereditary ocular developmental disorder that follows an autosomal dominant inheritance pattern. It is primarily caused by haploinsufficiency of the paired box gene 6 (PAX6) gene located on chromosome 11p13 [1,2]. PAX6 is a master regulatory transcription factor essential for ocular development, playing a pivotal role in the formation of multiple eye structures during embryogenesis [3]. In addition to the partial or complete absence of the iris, affected individuals often exhibit a spectrum of structural and functional ocular abnormalities, including lens subluxation or cataract, macular hypoplasia, optic nerve malformations, congenital or secondary glaucoma, and aniridia-associated keratopathy (AAK) [4,5]. Among these, limbal stem cell deficiency (LSCD) is recognized as a key pathological feature contributing to corneal opacification, neovascularization, and progressive vision loss in patients with AAK [6–8].

In patients with congenital aniridia, approximately 50% develop secondary glaucoma, which is commonly attributed to abnormalities in the development of the anterior chamber angle and Schlemm's canal, or to angle closure resulting from anterior rotation of the rudimentary iris [9–11]. In clinical practice, effective management of aniridia-associated glaucoma often necessitates the long-term use of prostaglandin analogues, such as travoprost, along with additional antiglaucoma medications to achieve adequate intraocular pressure control [12].

Travoprost is a highly selective and potent prostaglandin F2α analogue commonly used in the treatment of glaucoma and ocular hypertension. It exerts its effect

by selectively activating prostaglandin FP receptors, which leads to increased expression of matrix metalloproteinases (MMPs). These enzymes remodel the extracellular matrix of the ciliary muscle and sclera, thereby enhancing aqueous humor outflow through the uveoscleral pathway and effectively reducing intraocular pressure (IOP) [13–15]. Recent *in vivo* and *in vitro* studies suggest that, beyond its IOP-lowering properties, travoprost may also influence ocular surface tissues, particularly corneal epithelial cells. Reported effects include changes in cell viability, proliferation capacity, and cell–cell adhesion, along with elevated expression of inflammatory cytokines such as IL-6 and IL-8 [16–20]. However, there is a lack of systematic research on the specific mechanisms by which travoprost affects limbal epithelial cells, particularly in special populations such as patients with congenital aniridia.

Fos like 2 (FOSL2) is considered a target gene of PAX6 and is closely associated with corneal opacity [21]. It is also thought to play a role in regulating cell migration and proliferation [22]. Mitogen-activated protein kinases (MAPKs) are primarily categorized into three subfamilies: extracellular signal-regulated kinases (ERKs), c-Jun N-terminal kinases (JNKs), and mitogen-activated protein kinase 14 (MAPK14, also known as p38) [23]. Among these, ERKs and JNKs are particularly implicated in controlling cell proliferation, migration, and apoptosis [24–26]. As previously noted, travoprost can elevate MMP levels, thereby contributing to the reduction of IOP [27]. Related studies have also demonstrated that MMP expression is regulated through the JNK pathway [28]. In our earlier investigation of aniridia limbal stromal cells (AN-LSCs), we found that travoprost may influence MMP9 expression via the JNK signaling pathway, subsequently affecting cell migration [29]. Based on these findings, the present study focuses on evaluating the expression of these key genes and proteins.

Given its role as a first-line medication for glaucoma, the potential effects of travoprost on limbal epithelial cells (LECs)—particularly its mechanisms of action in special populations such as patients with congenital aniridia—warrant further investigation. Therefore, the aim of this study is to analyze changes in cell viability, proliferation, migration, as well as the expression of key genes and proteins associated with inflammation, apoptosis, and related pathways following travoprost treatment, using an siRNA-based aniridia LEC model, with PAX6 knockdown [30,31]. This work aims to evaluate the functional impact of travoprost on LECs and to provide theoretical insights and practical guidance for optimizing pharmacological treatment strategies in congenital aniridia patients with coexisting glaucoma.

## 2 Materials and Methods

### 2.1 Ethical approval and consent to participate

This study was approved by the Ethics Committee of Saarland, Germany (Approval Number: 124/23), and all procedures were conducted in accordance with the principles of the Declaration of Helsinki. Our research started on July 1st, 2023, and ended on January 31st, 2025.

### 2.2 Cell culture and PAX6 knockdown

Limbal biopsies were obtained from seven healthy corneal donors (mean age $77.1 \pm 10.0$ years; age range 60–91 years; 57.1% male) through the LIONS Cornea Bank Saar-Lor-Lux, Trier/Westpfalz, in Homburg/Saar, Germany. A summary of the donor characteristics is provided in Table 1. Using a 1.5 mm biopsy punch, tissue was collected from the limbal area of the cornea. The harvested samples were enzymatically digested in keratinocyte growth medium (KGM3; Promocell, Heidelberg, Germany) supplemented with collagenase A (Roche Diagnostic GmbH, Mannheim, Germany) at 37°C for 24 hours. Following digestion, LECs were seeded into single wells of six-well culture plates, with 2.5 mL of KGM3 per well. Upon reaching full confluence, the cells were detached using Trypsin-EDTA (Sigma-Aldrich, Saint Louis, USA) and subsequently reseeded into new six-well plates. The LECs were maintained under standard culture conditions at 37°C, with 95% humidity and 5% $CO_2$.

When LECs reached approximately 80% confluence, they were subjected to PAX6 gene silencing. For each well of a six-well plate, 2 mL of KGM3 medium was used in combination with a transfection mixture consisting of 500 µL of

**Table 1. Information on the corneal donors used.**

| Donor Number | Age (years) | Gender |
|---|---|---|
| 1 | 76 | Male |
| 2 | 72 | Male |
| 3 | 83 | Male |
| 4 | 91 | Female |
| 5 | 84 | Male |
| 6 | 60 | Female |
| 7 | 74 | Female |
| Total | 77.1 ± 10.0 (60-91) | 4 (57.1%) Male |

Opti-MEM™ I Reduced Serum Medium, GlutaMAX™ Supplement (Gibco, Thermo Fisher Scientific, Grand Island, USA), 5 µL of Lipofectamine™ 2000 (Thermo Fisher Scientific, Carlsbad, USA), and 1 µL of either Silencer™ Select PAX6 siRNA or Negative Control No.1 siRNA (Thermo Fisher Scientific, Waltham, USA).

Prior to application, Lipofectamine™ 2000 and siRNA were mixed and incubated for 20 minutes at room temperature in Opti-MEM™ to allow formation of the transfection complex. The experimental group received PAX6-targeting siRNA, while the control group was treated with non-targeting control siRNA. After 24 hours of transfection, the medium was replaced with 2.5 mL of fresh KGM3 medium to facilitate cellular recovery over the following 24 hours.

## 2.3 Travoprost treatment

Travoprost (Sigma-Aldrich, St. Louis, USA) was initially prepared in KGM3 medium at a concentration of 40 µg/mL (0.004%). This stock solution was subsequently twofold serially diluted to generate a concentration gradient ranging from 40 µg/mL to 0.039 µg/mL, specifically: 40, 20, 10, 5, 2.5, 1.25, 0.625, 0.313, 0.156, 0.078, and 0.039 µg/mL.

To assess cell viability and proliferation, experiments were conducted using 96-well plates. LECs were seeded in KGM3 medium, and upon reaching 80–90% confluence, the medium was replaced with the various concentrations of travoprost solutions, or with KGM3 alone as a negative control. After 20 minutes of exposure, the treatment solutions were removed, cells were rinsed with phosphate-buffered saline (PBS) (Merck, Sigma-Aldrich, Taufkirchen, Germany), and fresh KGM3 medium was added to each well.

For evaluation of cell migration and for RNA and protein extraction, the medium of both LECs and PAX6-knockdown LECs was replaced with KGM3 medium containing 0.156 µg/mL or 0.313 µg/mL of travoprost, or drug-free medium as a control. After a 20-minute incubation, the supernatant was collected, cells were rinsed with PBS, and fresh KGM3 medium was added to each well.

## 2.4 Cell viability assay

Cell viability was assessed using the Cell Proliferation Kit II (XTT) (Roche, Sigma-Aldrich, Mannheim, Germany) according to the manufacturer's instructions. Prior to use, 5 mL of XTT labeling solution was mixed with 0.1 mL of the electron coupling reagent, both of which were thawed and freshly prepared. A 50 µL aliquot of this working solution was added to each well of the 96-well plate. The cells were then incubated at 37 °C for 2 hours. Absorbance was measured at 450 nm, with a reference wavelength of 690 nm, using an Infinite F50 Absorbance Microplate Reader (TECAN Deutschland GmbH, Crailsheim, Germany).

## 2.5 Cell proliferation assay

Cell proliferation was assessed using the ELISA-BrdU (colorimetric) assay kit (Roche, Sigma-Aldrich, Mannheim, Germany). Briefly, cells were incubated with 10 µL of BrdU labeling reagent at 37 °C for 3 hours. Following incubation, the

labeling reagent was removed, and each well was treated with 200 µL of FixDenat solution for 30 minutes to fix and denature the DNA. Thereafter, a mixture of anti-BrdU-POD antibody and dilution buffer was added, followed by three washes with 200 µL of PBS to remove unbound antibodies. Upon the addition of the stop solution to the substrate, a colorimetric change was observed. Absorbance was measured using the Tecan Infinite F50 Absorbance Microplate Reader, as previously described.

## 2.6 Cell migration assay

To begin, three parallel reference lines were marked on the underside of each well in the six-well culture plates, spaced 5 mm apart. LECs were seeded into these wells with 2.5 mL of KGM3 medium. Following the previously described PAX6 knockdown protocol, a linear scratch was made perpendicular to the reference lines using 100 µL pipette tips (Eppendorf AG, Hamburg, Germany).

Cells were then treated for 20 minutes with either 0.156 µg/mL or 0.313 µg/mL of travoprost, or with travoprost-free medium as a control, as outlined earlier. Phase-contrast microscopy was used to capture images at baseline (0 hours) and at 6, 12, and 24 hours post-treatment. In each well, four separate wound areas aligned with the reference lines were imaged for subsequent analysis.

ImageJ software (https://imagej.nih.gov/ij/) was used to quantify wound closure. The migration rate (MR) was calculated using the following formula:

$$MR = [\text{Wound area } (0\,h) - \text{Wound area } (6/12/24h)] / \text{Wound area } (0\,h).$$

## 2.7 RNA/protein isolation and cDNA synthesis

Total RNA and protein were simultaneously isolated using the RNA/DNA/Protein Purification Plus Kit (Norgen Biotek, Ontario, Canada), following the manufacturer's instructions. The concentration of the extracted RNA was determined using a UV/VIS spectrophotometer (Analytik Jena AG, Jena, Germany), and samples were stored at –80 °C until further processing.

cDNA synthesis was carried out using the One Taq® RT-PCR Kit (New England Biolabs INC, Frankfurt, Germany), with 500 ng of total RNA used per reaction. The resulting cDNA was stored at –20 °C for subsequent applications.

## 2.8 Quantitative Real Time PCR

Primers used for quantitative real time Polymerase Chain Reaction (qRT-PCR) are listed in **Table 2**. Each qRT-PCR reaction was performed in a final volume of 9 µL, containing 1 µL of gene-specific primer mix, 5 µL of SYBR Green Master Mix (Vazyme, Nanjing, China), and 3 µL of nuclease-free water. In addition, 1 µL of synthesized cDNA was added, in accordance with the manufacturer's protocol. Reactions were carried out on a QuantStudio 5 Real-Time PCR System (Thermo Fisher Scientific, Waltham, USA) using the following thermal cycling conditions: initial denaturation at 95 °C for 10 seconds, followed by annealing/extension at 60 °C for 30 seconds, and a final denaturation step at 95 °C for 15 seconds, repeated for a total of 40 cycles.

All reactions were performed in duplicate. Gene expression levels were normalized against two housekeeping genes, TATA-binding protein (TBP) and β-glucuronidase (GUSB), using the comparative ΔΔCT method. Relative expression levels were calculated as fold changes (2^ΔΔCT), with untreated LECs serving as the baseline (fold change = 1).

## 2.9 Western blot analysis

For Western blot analysis, 20 µg of total protein per sample was mixed with 5 µL of Laemmli buffer, heated at 95 °C for 5 minutes, and loaded onto a 4–12% NuPAGE™ Bis-Tris SDS precast gel (Invitrogen, Waltham, MA, USA). To estimate

**Table 2. Primers used for quantitative real time PCR (qRT-PCR).**

| Primer | Gene Symbol | QIAGEN catalog number | Manufacturer (company, city, country) |
|---|---|---|---|
| Caspase-3 | CASP3 | QT00023947 | Qiagen N.V., Venlo, Netherlands |
| FOSL2 | FOSL2 | QT01000881 | Qiagen N.V., Venlo, Netherlands |
| GUSB | GUSB | QT00046046 | Qiagen N.V., Venlo, Netherlands |
| IL-1α | IL1A | QT00001127 | Qiagen N.V., Venlo, Netherlands |
| IL-1β | IL1B | QT00021385 | Qiagen N.V., Venlo, Netherlands |
| IL-6 | IL6 | QT00083720 | Qiagen N.V., Venlo, Netherlands |
| IL-8 | CXCL8 | QT00000322 | Qiagen N.V., Venlo, Netherlands |
| MAPK1 | MAPK1 | QT00065933 | Qiagen N.V., Venlo, Netherlands |
| MAPK3 | MAPK3 | QT02589314 | Qiagen N.V., Venlo, Netherlands |
| MAPK8 | MAPK8 | QT00091056 | Qiagen N.V., Venlo, Netherlands |
| MMP9 | MMP9 | QT00040040 | Qiagen N.V., Venlo, Netherlands |
| NF-κB | RELA | QT02324308 | Qiagen N.V., Venlo, Netherlands |
| PAX6 | PAX6 | QT00071169 | Qiagen N.V., Venlo, Netherlands |
| PTGES2 | PTGES2 | QT00082068 | Qiagen N.V., Venlo, Netherlands |
| TBP | TBP | QT00000721 | Qiagen N.V., Venlo, Netherlands |
| TNFα | TNF | QT00029162 | Qiagen N.V., Venlo, Netherlands |

molecular weight, 2.5 μL of Precision Plus Protein™ All Blue Prestained Protein Standards (Bio-Rad Laboratories, Hercules, USA) was loaded into the first lane. Electrophoresis was performed using NuPAGE™ MOPS SDS Running Buffer (20×) (Thermo Fisher Scientific, Waltham, USA).

Following electrophoretic separation, proteins were transferred to a nitrocellulose membrane using the Trans-Blot Turbo Transfer System (Bio-Rad Laboratories) according to the manufacturer's pre-set protocol optimized for high molecular weight proteins via semi-dry transfer. Total protein normalization was performed using No-Stain™ Protein Labeling Reagent (Thermo Fisher Scientific, Dreieich, Germany). Membranes were washed three times with 10 mL of Western Froxx washing buffer (BioFroxx GmbH, Einhausen, Germany) for 5 minutes each.

Membranes were then incubated overnight at 4 °C with primary antibodies listed in **Table 3**, diluted in a combined blocking and secondary antibody solution containing WesternFroxx anti-Rabbit or anti-Mouse HRP (BioFroxx GmbH). Following incubation, three additional washes were conducted to remove unbound antibodies. Protein detection was carried out using Western Lightning Plus ECL substrate (PerkinElmer Inc., Waltham, USA), and chemiluminescent signals were captured using the iBright™ CL1500 Imaging System (Thermo Fisher Scientific, Waltham, USA). After detection, membranes were treated with Western Froxx stripping solution (BioFroxx GmbH) to allow reprobing with alternative antibodies.

## 2.10 Enzyme linked immunosorbent assay (ELISA)

Quantification of IL-1α, IL-1β, IL-6, IL-8, TNFα, and MMP9 protein levels was performed using DuoSet® ELISA Kits (R&D Systems, Bio-Techne, Minneapolis, USA), as detailed in **Table 4** and in accordance with the manufacturer's instructions. Each sample was analyzed in duplicate, and absorbance was measured at 450 nm using the Tecan Infinite F50 Absorbance Microplate Reader. The resulting cytokine concentrations were normalized to total protein content of the supernatant to obtain final values.

## 2.11 Statistical analysis

Statistical analysis and graph generation were conducted using GraphPad Prism software (version 9.2.0). Results from cell viability, proliferation, and migration experiments are presented as mean±standard deviation (SD), while qRT-PCR

**Table 3. Antibodies used for Western blot analysis.**

| Antibody | Dilution | Catalog number | Manufacturer |
|---|---|---|---|
| Caspase3/p17/p19 Polyclonal antibody | 1:1000 | 19677-1-AP | Proteintech Group, Rosemont, USA |
| FRA2 (FOSL2) Mouse Monoclonal Antibody | 1:2000 | TA809660 | OriGene Technologies, Inc., Rockville, USA |
| p44/42 MAPK (Erk1/2) (137F5) Rabbit mAb | 1:1000 | 4695 | Cell Signaling Technology, Danvers, USA |
| JNK Polyclonal antibody | 1:5000 | 24164-1-AP | Proteintech Group, Rosemont, USA |
| NF-κB p65 (D14E12) XP Rabbit mAb | 1:1000 | 8242 | Cell Signaling Technology, Danvers, USA |
| PAX6 Rabbit Polyclonal antibody | 1:500 | 12323-1-AP | Proteintech Group, Rosemont, USA |
| PTGES2 Polyclonal antibody | 1:500 | 10881-1-AP | Proteintech Group, Rosemont, USA |

**Table 4. ELISA kits used in the study.**

| ELISA kit | Catalog number | Manufacturer |
|---|---|---|
| Human IL-1 alpha/IL-1F1 DuoSet ELISA | DY-200–05 | R&D Systems, Bio-Techne, Minneapolis, USA |
| Human IL-1 beta/IL-1F2 DuoSet ELISA | DY-201–05 | R&D Systems, Bio-Techne, Minneapolis, USA |
| Human IL-6 DuoSet ELISA | DY206 | R&D Systems, Bio-Techne, Minneapolis, USA |
| Human IL-8/CXCL8 DuoSet ELISA | DY208 | R&D Systems, Bio-Techne, Minneapolis, USA |
| Human MMP-9 DuoSet ELISA | DY911 | R&D Systems, Bio-Techne, Minneapolis, USA |
| Human TNF-alpha DuoSet ELISA | DY210−05 | R&D Systems, Bio-Techne, Minneapolis, USA |

data are expressed as geometric mean±geometric SD. Western blot results are also reported as mean±SD. One-way ANOVA was used to assess differences in cell viability and proliferation. For comparisons involving cell migration, gene expression ($2^{\Delta\Delta Ct}$), and protein quantification across varying concentrations of travoprost, two-way ANOVA followed by Tukey's post hoc test was applied. A p-value <0.05 was considered statistically significant.

## 3 Results

### 3.1 Cell viability and proliferation

The XTT assay and BrdU assay were employed to assess LECs viability and proliferation, respectively. Changes in LECs viability and proliferation following 20-minute treatments with varying concentrations of travoprost are summarized in **Fig 1** and **Fig 2**. Regarding cell viability, a significant decrease was observed starting at a travoprost concentration of 0.156 μg/mL compared to the control group (p = 0.028).

For cell proliferation, a significant reduction was noted in LECs treated with 20 μg/mL and 40 μg/mL of travoprost for 20 minutes (p = 0.029 and p = 0.004, respectively). No significant changes in proliferation were observed at lower concentrations (p ≥ 0.128).

Based on these results, travoprost concentrations of 0.156 μg/mL and 0.313 μg/mL were used for the following experiments.

### 3.2 Cell migration

The scratch assay was used to evaluate cell migration ability. **Fig 3** presents representative scratch assay images at 0, 6, 12, and 24 hours after travoprost treatment, along with the corresponding migration rates of LECs and PAX6-knockdown LECs at 6, 12, and 24 hours. At all three time points—6, 12, and 24 hours post-treatment—PAX6-knockdown LECs demonstrated significantly lower migration rates compared to LECs (p = 0.001, p = 0.046, and p < 0.001, respectively). At 6 hours, the migration rate of untreated PAX6-knockdown LECs was significantly reduced compared to that of untreated LECs (p = 0.002). Furthermore, in PAX6-knockdown LECs, treatment with 0.313 μg/mL travoprost led to a

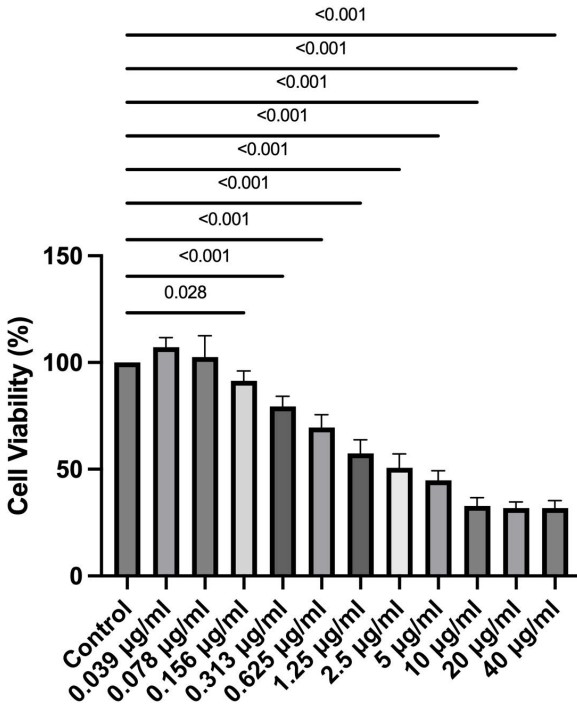

**Fig 1. Cell viability of primary human limbal epithelial cells (LECs) after 20-minute treatment with travoprost at concentrations ranging from 0.039 μg/mL to 40 μg/mL (n = 7).** A significant reduction in LEC viability was observed starting at a travoprost concentration of 0.156 μg/mL (p ≤ 0.028).

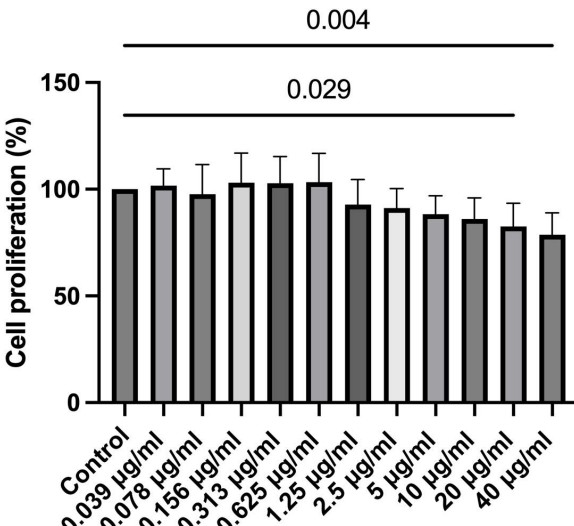

**Fig 2. Cell proliferation of limbal epithelial cells (LECs) after 20-minute treatment with travoprost at concentrations ranging from 0.039 μg/mL to 40 μg/mL, using the ELISA-BrdU (colorimetric) kit (n = 7).** LEC proliferation decreased significantly following treatment with 20 μg/mL and 40 μg/mL concentrations of travoprost (p = 0.029 and p = 0.006, respectively).

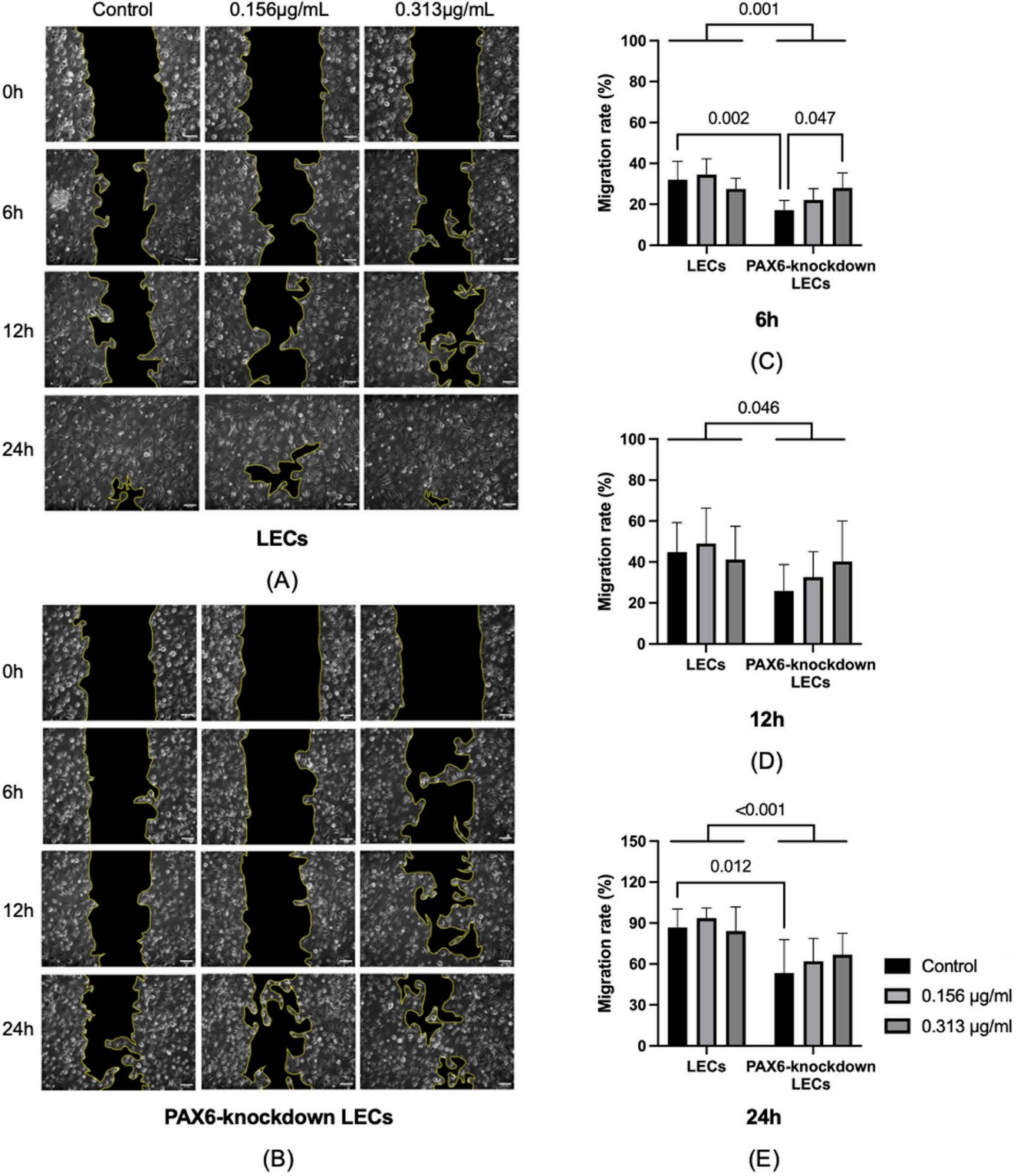

**Fig 3. Images of scratch assays (A, B) and quantification of migration rates (C, D, E) for limbal epithelial cells (LECs) and PAX6-knockdown LECs at 0, 6, 12, and 24 hours following a 20-minute treatment with travoprost at concentrations of 0.156 μg/mL and 0.313 μg/mL (n = 5).**
At 6, 12, and 24 hours, the migration rate in LECs was significantly higher than in PAX6-knockdown LECs (p ≤ 0.046). At 6 hours, untreated LECs

demonstrated a significantly higher migration rate compared to untreated PAX6-knockdown LECs (p = 0.002) **(C)**. Additionally, PAX6-knockdown LECs treated with 0.313 μg/mL travoprost showed a significantly higher migration rate than untreated PAX6-knockdown cells (p = 0.047). At 24 hours, untreated LECs continued to exhibit a significantly higher migration rate than their PAX6-knockdown counterparts (p = 0.012) **(E)**. **Scale bar: 50 μm.**

significantly higher migration rate than in the untreated group (p = 0.047). At 24 hours, this trend persisted, with untreated PAX6-knockdown LECs continuing to show significantly lower migration than untreated LECs (p = 0.012).

### 3.3 PAX6 mRNA and protein levels

We compared the mRNA and protein levels of PAX6 in LECs and PAX6-knockdown LECs to verify the success of the PAX6 knockdown. Compared to LECs, PAX6-knockdown LECs showed a significant reduction in *PAX6* mRNA levels (p < 0.001). Similarly, PAX6 protein levels were also significantly lower in the knockdown group (p = 0.004). However, travoprost treatment had no significant effect on PAX6 mRNA or protein level in either cell type (p ≥ 0.773) (**Fig 4**).

### 3.4 *Ki67* mRNA levels

Ki67 is widely regarded as a marker of cell proliferation [32]. The mRNA levels of *Ki67* did not differ significantly between LECs and PAX6-knockdown LECs (p = 0.321). Moreover, treatment with travoprost did not affect *Ki67* mRNA expression in either the LEC or PAX6-knockdown subgroups (p ≥ 0.867) (**Fig 5**).

### 3.5 FOSL2 and mitogen-activated protein kinases (MAPKs)

FOSL2 is regarded as a downstream gene regulated by PAX6 and has been closely linked to corneal opacity. It is also thought to contribute to the control of cell migration and proliferation [21,22]. MAPKs are similarly recognized for their roles in governing key cellular processes, including proliferation, migration, and apoptosis [24–26]. In PAX6-knockdown LECs, the mRNA levels of *FOSL2*, *ERK1 (MAPK3)*, *ERK2 (MAPK1)*, and *JNK (MAPK8)* were significantly lower than those observed in LECs (p = 0.007, p < 0.001, p = 0.016, p = 0.043). In addition, the protein levels of JNK1/2 were significantly reduced in untreated PAX6-knockdown LECs compared to untreated LECs (p = 0.046). Treatment with 0.313 μg/mL travoprost significantly increased JNK1/2 protein expression in PAX6-knockdown LECs (p = 0.039) (**Fig 6**).

### 3.6 Inflammation-related genes and proteins

In PAX6-knockdown LECs, the mRNA levels of inflammation-related genes *IL-1α*, *IL-1β*, *IL-6*, *IL-8*, *TNFα*, and *NF-κB* were significantly lower than those in LECs (p = 0.019, p = 0.015, p < 0.001, p = 0.031, p < 0.001, p = 0.017, respectively). There were no significant differences in *PTGES2* mRNA levels between LECs and PAX6-knockdown LECs, nor among different concentrations of travoprost (p ≥ 0.175). At the protein level, IL-6 was significantly reduced in PAX6-knockdown LECs compared to LECs (p = 0.025), while travoprost treatment had no significant effect on IL-6 expression (p ≥ 0.589). Additionally, no significant differences were found in the protein levels of IL-1α, IL-1β, IL-8, TNFα, NF-κB, and PTGES2 between LECs and PAX6-knockdown LECs, or across the different travoprost concentrations (p ≥ 0.155) (**Fig 7**).

### 3.7 Caspase-3 mRNA and protein levels

Caspase-3 is a crucial enzyme involved in the apoptotic process and is commonly used as a marker of apoptosis [33]. In PAX6-knockdown LECs, the mRNA level of *caspase-3* was significantly lower compared to that in LECs (p = 0.017). However, when treated with 0.313 μg/mL travoprost, PAX6-knockdown LECs exhibited significantly higher caspase-3 protein levels than LECs treated with the same concentration (p = 0.019). Furthermore, in PAX6-knockdown LECs, 0.313 μg/mL travoprost significantly increased caspase-3 protein level compared to the untreated group (p = 0.044) (**Fig 8**).

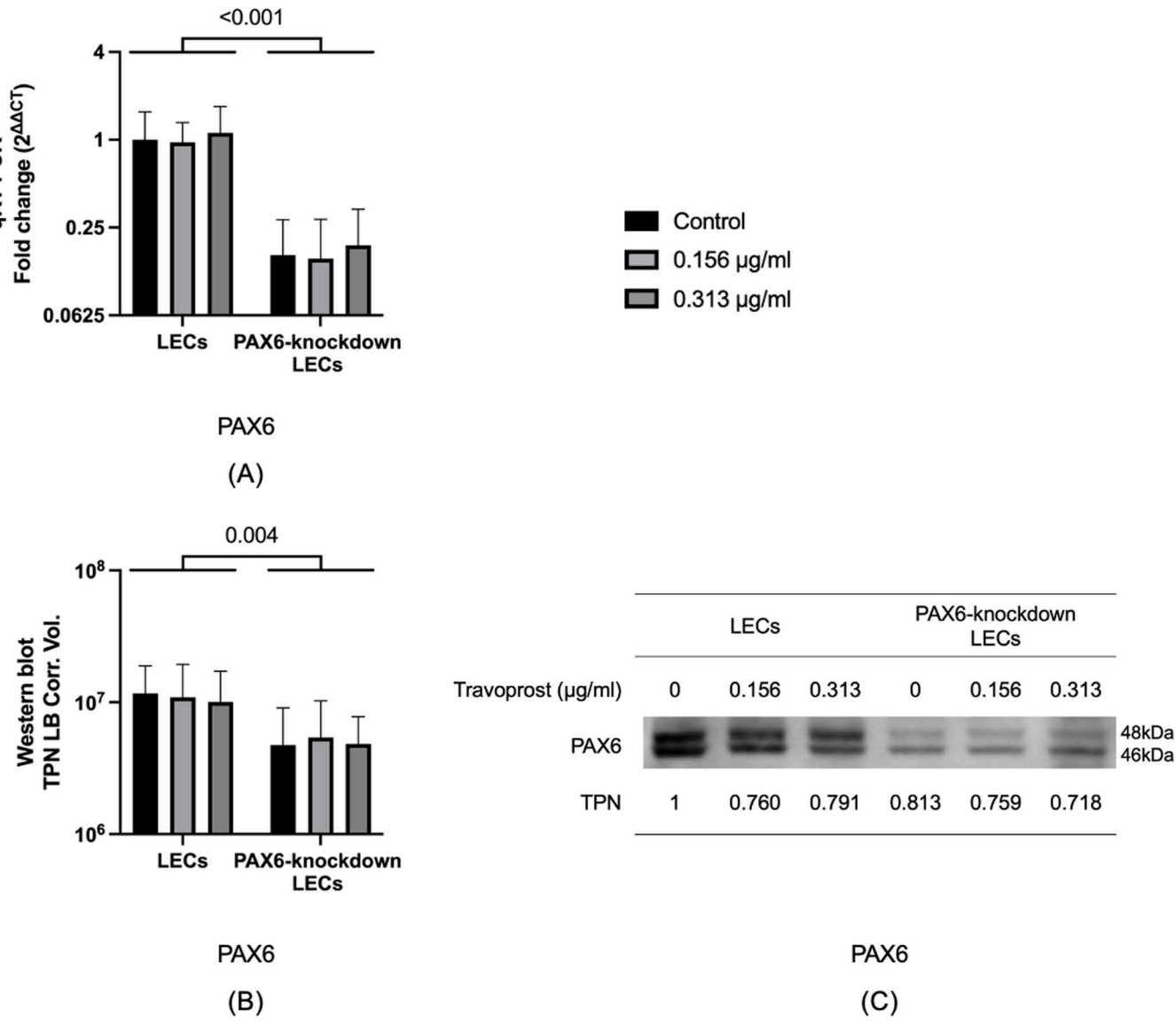

**Fig 4. PAX6 mRNA levels, protein levels, and representative Western blot in limbal epithelial cells (LECs) and PAX6-knockdown LECs following a 20-minute treatment with travoprost at concentrations of 0.156 µg/mL and 0.313 µg/mL (n=7) (A-C).** mRNA values are presented on a logarithmic scale (Log$_2$) and expressed as the geometric mean±geometric standard deviation (SD). Protein levels are shown on a logarithmic scale (Log$_{10}$) as mean±SD. Two-way ANOVA was used for statistical analysis, and significant p-values (p<0.05) are indicated in the diagrams. TPN LB Corr. Vol. refers to the total lane protein-normalized, local background-corrected volume on the Y-axis. *PAX6* mRNA levels were significantly lower in PAX6-knockdown LECs compared to LECs (p<0.001) **(A)**. Similarly, PAX6 protein levels were significantly reduced in PAX6-knockdown LECs relative to LECs (p=0.004) **(B)**.

## 3.8 MMP9 mRNA and protein levels

An increase in MMP9 expression is one of the mechanisms through which travoprost reduces IOP and is also linked to the regulation of cell migration [27,34]. In PAX6-knockdown LECs, *MMP9* mRNA levels were significantly higher than in LECs (p=0.003). Furthermore, in PAX6-knockdown LECs treated with 0.313 µg/mL travoprost, *MMP9* mRNA levels were significantly higher than those in LECs treated with the same concentration (p=0.010). Travoprost treatment at

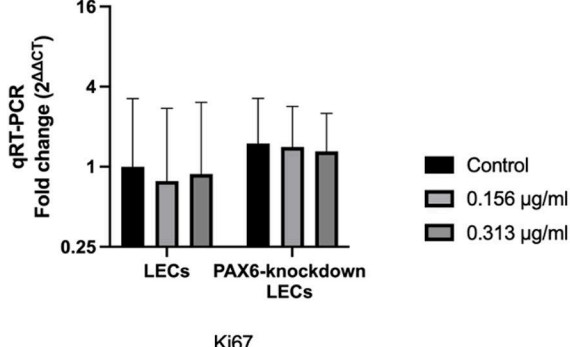

**Fig 5. *Ki67* mRNA levels in limbal epithelial cells (LECs) and PAX6-knockdown LECs following a 20-minute treatment with travoprost at concentrations of 0.156 µg/mL and 0.313 µg/mL (n = 7).** mRNA values are presented on a logarithmic scale (Log$_2$) and expressed as the geometric mean±geometric standard deviation (SD). Two-way ANOVA was used for statistical analysis, and significant p-values (p < 0.05) are indicated in the diagrams. No significant differences were observed between any of the analyzed groups or subgroups (p ≥ 0.321).

0.313 µg/mL also significantly upregulated *MMP9* mRNA level in PAX6-knockdown LECs compared to their untreated counterparts (p = 0.021).

Similarly, MMP9 protein levels were significantly elevated in PAX6-knockdown LECs compared to LECs (p < 0.001). In addition, treatment with 0.156 µg/mL and 0.313 µg/mL travoprost further increased MMP9 protein level in PAX6-knockdown LECs (p = 0.044 and p = 0.027, respectively) (**Fig 9**).

## 4 Discussion

Congenital aniridia is a complex hereditary ocular developmental disorder, primarily caused by mutations in the PAX6 gene, resulting in reduced PAX6 gene and protein expression. This deficiency leads to widespread abnormalities of the ocular surface, particularly the development of AAK [35,36]. In this study, we used an siRNA-mediated PAX6 knockdown model in limbal epithelial cells and, for the first time, systematically investigated the effects of varying concentrations of the prostaglandin analogue travoprost on both normal LECs and PAX6-knockdown LECs. Our evaluations included analyses of cell viability, proliferation, and migration, along with the expression of key genes and proteins such as FOSL2, MAPKs, inflammatory markers, caspase-3, and MMP9. The findings demonstrated that PAX6-knockdown LECs were more sensitive to travoprost than normal LECs, and offer important experimental evidence to develop pharmacological strategies for treating patients with congenital aniridia complicated by glaucoma.

Our experimental data demonstrated that travoprost significantly reduced the viability of LECs at concentrations as low as 0.156 µg/mL, with a clear dose-dependent effect as the concentration increased. At lower concentrations, the reduction in LEC viability induced by travoprost may not result from overt cytotoxicity but rather from the early activation of stress- and apoptosis-related signaling pathways. In PAX6-knockdown LECs, we observed an increase in caspase-3 protein levels following treatment with 0.313 µg/mL travoprost, supporting the hypothesis that travoprost can trigger apoptotic processes even at relatively low concentrations. This early response appears to be more pronounced in PAX6-deficient cells, which exhibit impaired regulatory capacity and heightened sensitivity to external stimuli. Similarly, we found that low-concentration travoprost treatment elevated JNK protein levels in PAX6-knockdown LECs, suggesting that travoprost may also induce mild oxidative stress under these conditions [37]. This finding aligns with previous studies on corneal epithelial cell lines [17,19], where exposure to 40 µg/mL travoprost for 5, 15, or 30 minutes resulted in significantly decreased cell viability. Notably, our study examined a broader range of travoprost concentrations, allowing for a more detailed assessment of its impact on LEC viability. Interestingly, while LEC viability was reduced even at low concentrations (0.156

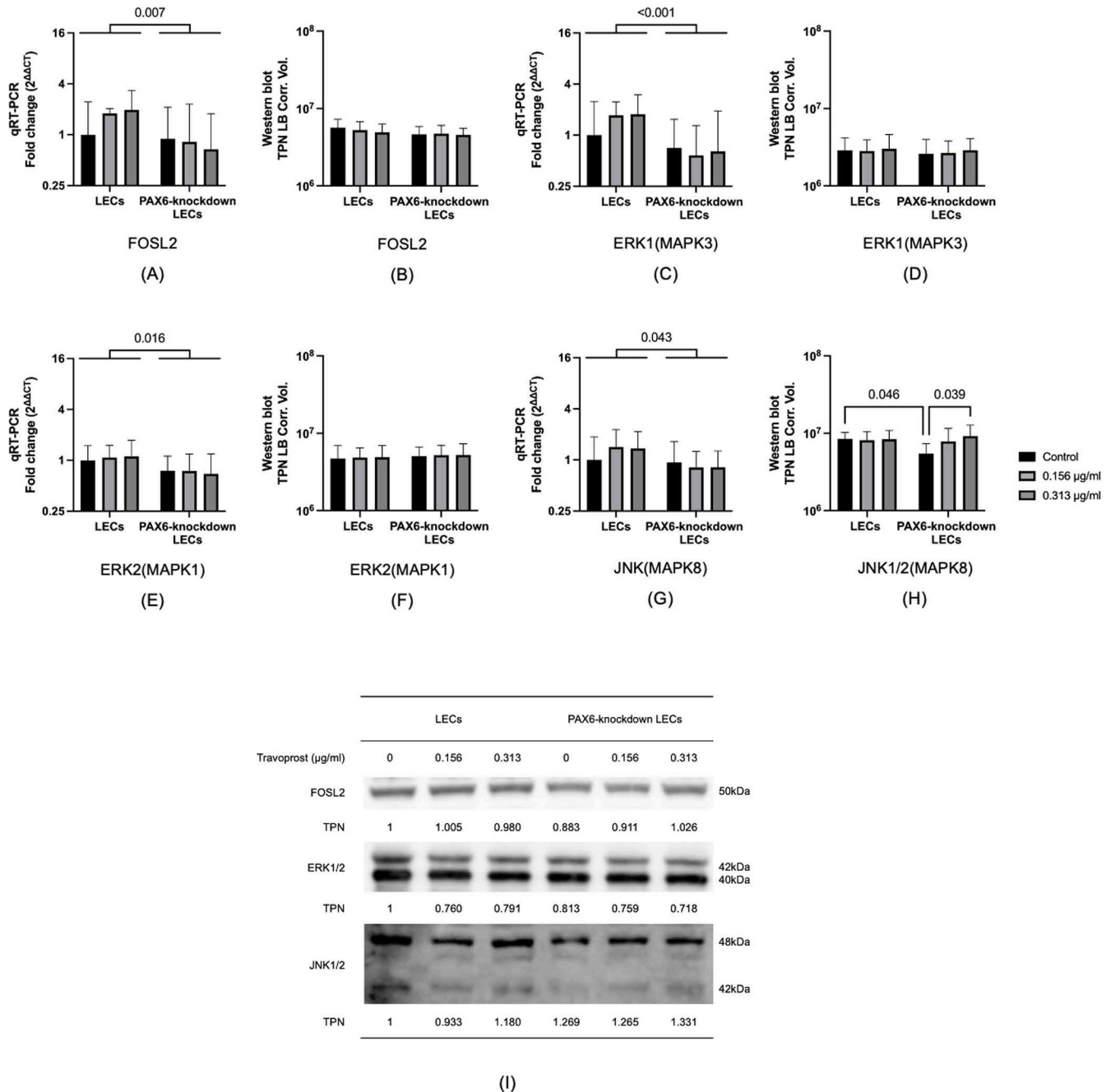

**Fig 6. FOSL2, ERK1 (MAPK3), ERK2 (MAPK1), and JNK (MAPK8) mRNA and protein levels, and representative Western blots for FOSL2, ERK1/2, and JNK1/2 in limbal epithelial cells (LECs) and PAX6-knockdown LECs following a 20-minute treatment with travoprost at concentrations of 0.156 µg/mL and 0.313 µg/mL (n = 7) (A–I).** mRNA values are presented on a logarithmic scale (Log$_2$) and expressed as the geometric mean±geometric standard deviation (SD). Protein levels are shown on a logarithmic scale (Log$_{10}$) as mean±SD. Two-way ANOVA was used for statistical analysis, and significant p-values (p < 0.05) are indicated in the diagrams. TPN LB Corr. Vol. refers to the total lane protein-normalized, local background-corrected volume on the Y-axis. **mRNA levels of *FOSL2, ERK1, ERK2,* and *JNK* were significantly lower in PAX6-knockdown LECs compared to LECs (p ≤ 0.043) (A, C, E, G).** JNK1/2 protein levels were significantly reduced in untreated PAX6-knockdown LECs compared to LECs (p = 0.046), while treatment with 0.313 µg/mL travoprost significantly increased JNK1/2 protein levels in PAX6-knockdown LECs compared to untreated PAX6-knockdown cells (p = 0.039) (**H**).

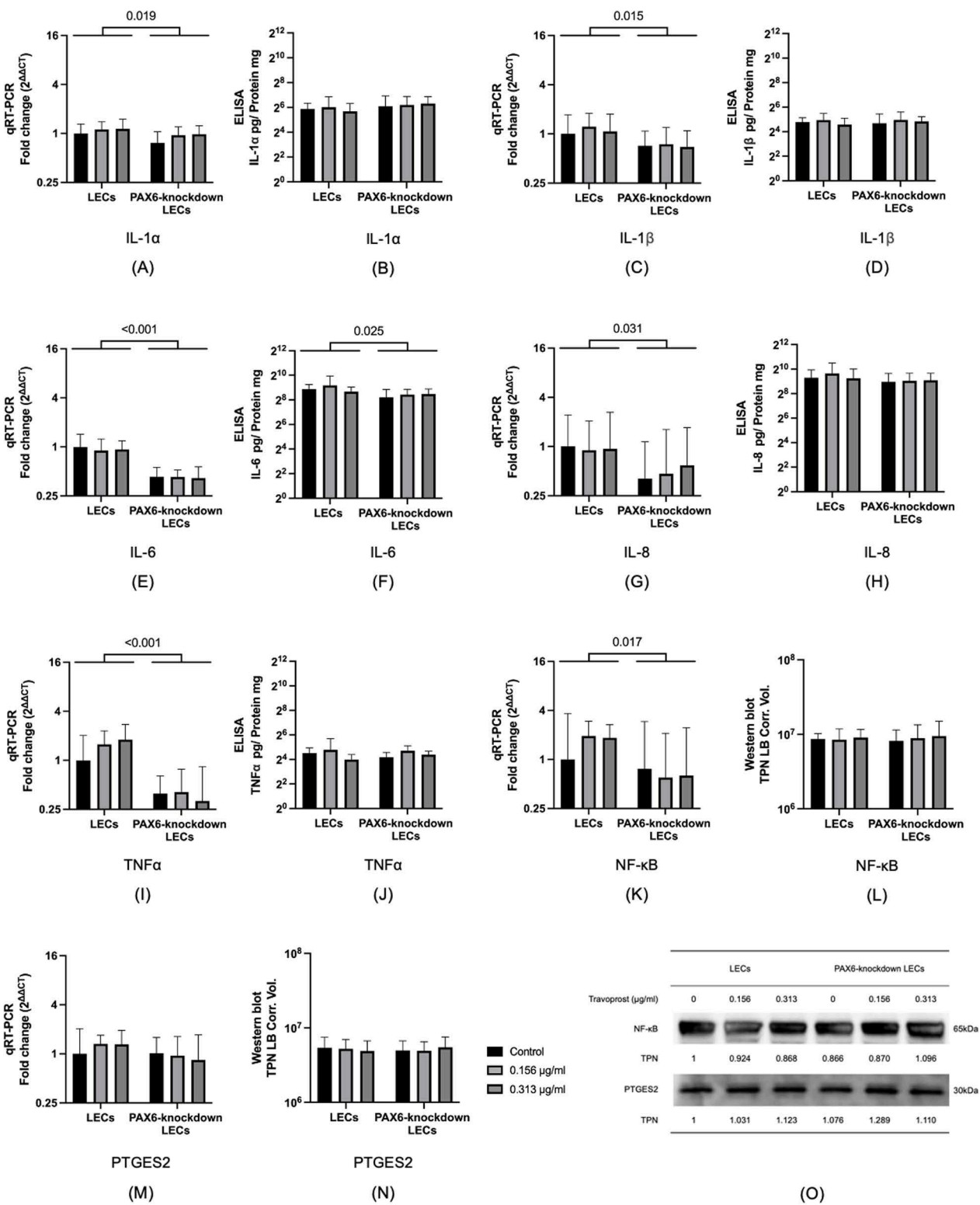

**Fig 7. mRNA and protein levels of inflammation-related markers including IL-1α, IL-1β, IL-6, IL-8, TNFα, NF-κB, and PTGES2, as well as representative Western blots for each marker, in limbal epithelial cells (LECs) and PAX6-knockdown LECs following a 20-minute treatment**

with travoprost at concentrations of 0.156 μg/mL and 0.313 μg/mL (n = 7) (A-O). mRNA expression values are presented on a logarithmic scale (Log$_2$) as geometric mean±geometric standard deviation (SD). Protein levels of IL-1α, IL-1β, IL-6, IL-8, and TNFα were determined by Western blot and are shown on a logarithmic scale (Log$_{10}$) as mean±SD. NF-κB and PTGES2 protein levels were measured using ELISA, also presented on a Log$_{10}$ scale as mean±SD. Statistical analysis was performed using two-way ANOVA, and p-values <0.05 are indicated in the figures. TPN LB Corr. Vol. refers to the total lane protein-normalized, local background-corrected volume plotted on the Y-axis. *IL-1α, IL-1β, IL-6, IL-8, TNFα*, and *NF-κB* mRNA levels were significantly lower in PAX6-knockdown LECs compared to LECs (p ≤ 0.031) (A, C, E, G, I, K). The IL-6 protein level was also significantly reduced in PAX6-knockdown LECs relative to LECs (p = 0.025) (F). No significant differences were observed between any of the other analyzed groups or subgroups.

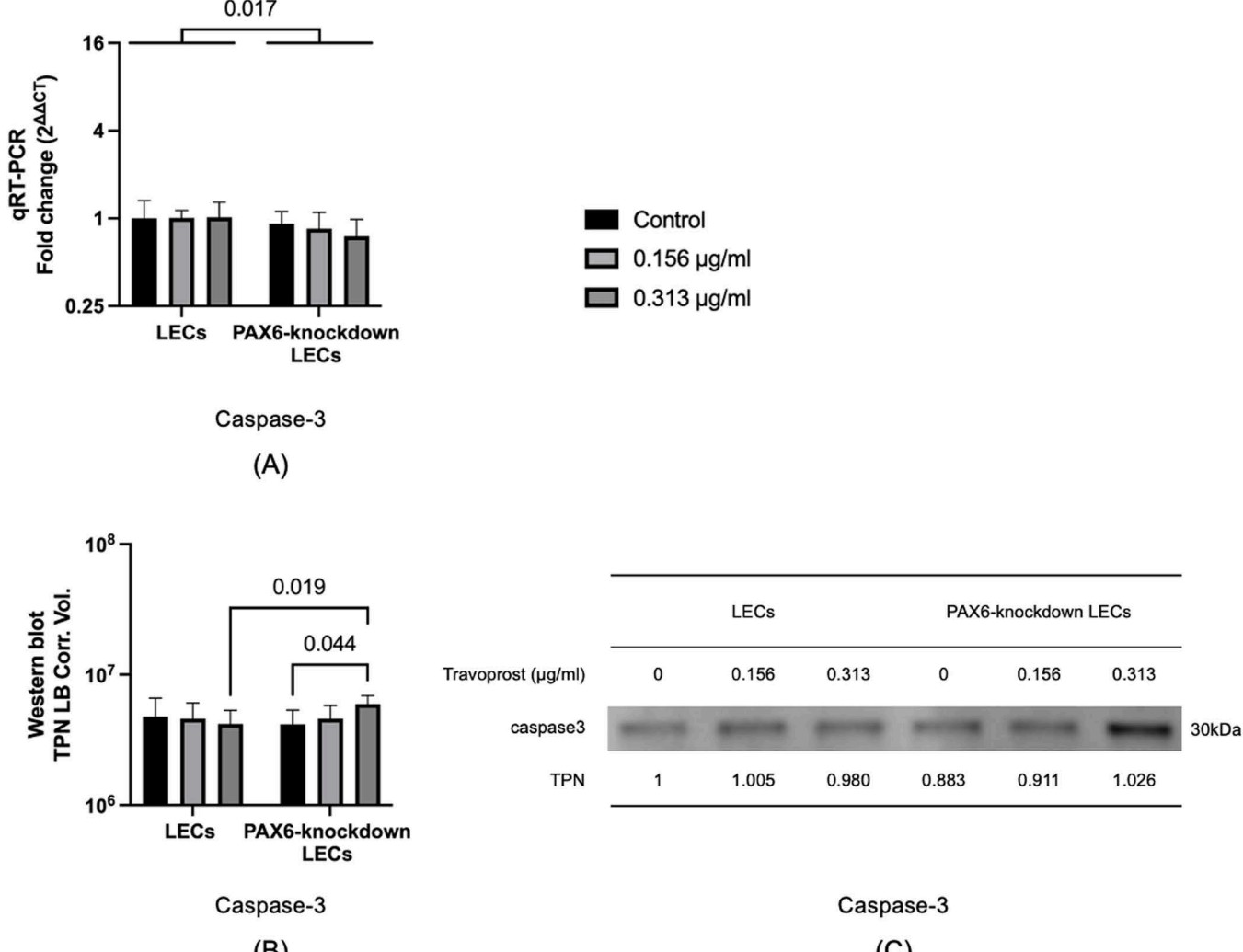

**Fig 8. Caspase-3 mRNA and protein levels, and representative Western blot in limbal epithelial cells (LECs) and PAX6-knockdown LECs following a 20-minute treatment with travoprost at concentrations of 0.156 μg/mL and 0.313 μg/mL (n = 7) (A-C).** mRNA values are presented on a logarithmic scale (Log$_2$) as geometric mean±geometric standard deviation (SD). Protein values are shown on a logarithmic scale (Log$_{10}$) as mean±SD. Two-way ANOVA was used for statistical analysis, and significant p-values (p < 0.05) are indicated in the diagrams. TPN LB Corr. Vol. refers to the total lane protein–normalized, local background–corrected volume displayed on the Y-axis. ***Caspase-3*** mRNA levels were significantly lower in PAX6-knockdown LECs than in LECs (**p** = 0.017) (A). Caspase-3 protein levels were significantly higher in PAX6-knockdown LECs treated with 0.313 μg/mL travoprost compared to both LECs treated with the same concentration (p = 0.019) and untreated PAX6-knockdown LECs (p = 0.044). (B).

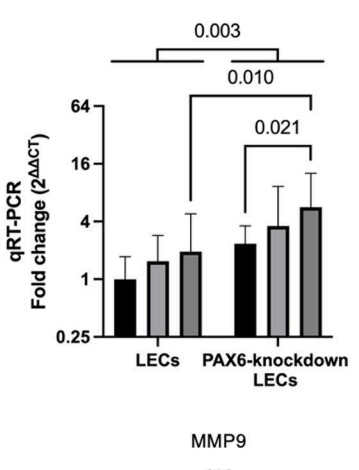
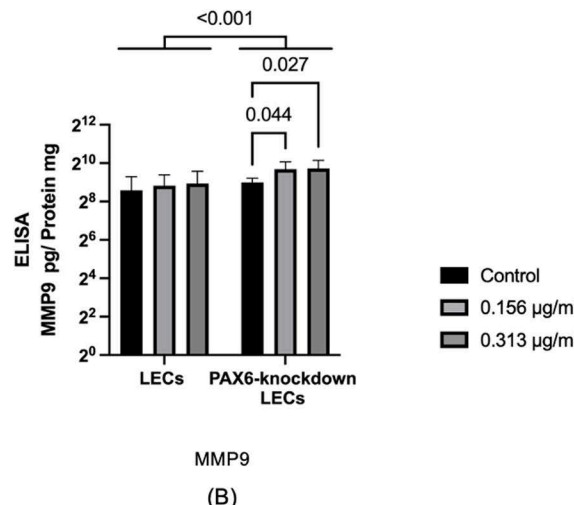

MMP9
(A)

MMP9
(B)

**Fig 9. MMP9 mRNA and protein levels in limbal epithelial cells (LECs) and PAX6-knockdown LECs following a 20-minute treatment with travoprost at concentrations of 0.156 μg/mL and 0.313 μg/mL (n = 7) (A, B).** mRNA values are presented on a logarithmic scale ($Log_2$) as geometric mean±geometric standard deviation (SD). Protein values are also shown on a logarithmic scale ($Log_2$) as mean±SD. Statistical analysis was performed using two-way ANOVA, and significant p-values ($p < 0.05$) are indicated in the diagrams. *MMP9* mRNA levels were significantly higher in PAX6-knockdown LECs than in LECs (**p** = 0.003). Additionally, *MMP9* mRNA levels were significantly elevated in PAX6-knockdown LECs treated with 0.313 μg/mL travoprost compared to both travoprost-treated LECs (p = 0.010) and untreated PAX6-knockdown LECs (p = 0.021) **(A)**. **MMP9 protein levels were significantly higher in PAX6-knockdown LECs compared to LECs (p < 0.001). Treatment with 0.156 μg/mL and 0.313 μg/mL travoprost significantly increased MMP9 protein levels in PAX6-knockdown LECs compared to untreated controls (p = 0.044 and p = 0.027, respectively) (B).**

μg/mL), proliferation was only suppressed at higher concentrations (20 μg/mL and 40 μg/mL). This is consistent with findings by Liang et al. [38], who reported that the proliferation of immortalized human corneal epithelial cells was significantly reduced after a 30-minute exposure to 40 μg/mL travoprost. Importantly, our study excluded the effects of preservatives commonly found in antiglaucomatous eye drops, allowing us to assess the direct effects of travoprost itself. We also examined the effects of 0.156 and 0.313 μg/mL travoprost on Ki67 mRNA levels, a marker of cell proliferation [32], in both LECs and PAX6-knockdown LECs. These travoprost concentrations did not affect proliferation in either group, consistent with our BrdU assay results.

In the cell migration assay, PAX6-knockdown LECs exhibited a significantly lower migration rate compared to normal LECs. However, treatment with 0.313 μg/mL travoprost notably enhanced migration in the knockdown cells, suggesting a potential "restorative" effect that may be mediated through multiple mechanisms.

First, FOSL2, a known transcriptional target of PAX6, has been implicated in maintaining corneal transparency [21]. In our study, *FOSL2* mRNA levels were significantly reduced following PAX6 knockdown. As a member of the activator protein-1 (AP-1) transcription factor family, FOSL2 is involved in the regulation of cell proliferation and migration [39–41]. Therefore, the impaired migration observed in PAX6-deficient LECs may, in part, be attributed to the downregulation of *FOSL2*.

Second, our MAPK pathway analysis showed that PAX6 knockdown resulted in downregulation of *ERK1/2* and *JNK* mRNA levels. Interestingly, travoprost treatment led to an upregulation of JNK1/2 protein levels in PAX6-knockdown LECs. Given that JNK signaling plays a critical role in regulating cell migration [42,43], this suggests that travoprost may partially restore migratory capacity in PAX6-deficient cells through JNK activation.

Furthermore, our results demonstrated that travoprost upregulated both the gene and protein expression of MMP9 in PAX6-knockdown LECs. The upregulation of matrix metalloproteinases (MMPs) and the resulting extracellular matrix

(ECM) remodeling is a well-established mechanism by which prostaglandin analogues, including travoprost, facilitate intraocular pressure reduction [27,44]. MMP9, in particular, has been shown to promote epithelial cell migration [34], and its expression is known to be regulated by the JNK signaling pathway [45,46]. Taken together, these findings suggest that travoprost may enhance the migratory capacity of PAX6-knockdown LECs via the JNK–MMP9 axis. However, it is important to consider that excessive MMP expression could potentially disrupt ECM homeostasis in corneal tissue, and its role in the progression of aniridia-associated keratopathy remains an open question that warrants further investigation.

Regarding the expression of inflammation-related genes and proteins, PAX6 knockdown led to a reduction in mRNA levels for most inflammatory markers. This effect may be attributed to the suppression of the JNK/ERK signaling pathways, thereby attenuating the inflammatory response [47]. Additionally, treatment with 0.156 µg/mL and 0.313 µg/mL travoprost did not elicit any significant changes in inflammatory marker expression in either normal LECs or PAX6-knockdown LECs. Unlike several previous *in vivo* and *in vitro* studies on travoprost [19,48], our study was specifically designed to assess the inflammatory effects of the active compound alone, intentionally excluding potential confounding effects from preservatives. In those earlier studies, travoprost was reported to increase the expression of inflammatory cytokines such as IL-6, IL-8, and TNF-α in corneal cells. However, it seems that these pro-inflammatory effects were at least partially driven by the presence of preservatives, whose detrimental effects on corneal cells have been well documented [19,49,50].

Caspase-3 is recognized as a key executioner enzyme in the apoptotic process, exhibiting proteolytic activity and regulating the expression of downstream effectors in both the intrinsic and extrinsic apoptotic pathways, ultimately leading to programmed cell death [51]. In our study, treatment with 0.313 µg/mL travoprost led to a significant increase in caspase-3 protein expression in PAX6-knockdown LECs. This observation is consistent with findings by Paimela et al. [19], where treatment of corneal epithelial cell lines with 40 µg/mL travoprost for 5, 15, and 30 minutes also resulted in an upward trend in caspase-3 expression. However, it should be emphasized that the potential pro-apoptotic effects of preservatives in those studies cannot be ruled out. Moreover, the JNK signaling pathway, which is known to be involved in apoptotic regulation, may also play a role in modulating caspase-3 expression. The travoprost-induced activation of JNK, as observed in our study, could contribute to the upregulation of caspase-3 and provide a possible mechanistic link between PAX6 deficiency, JNK activation, and apoptotic susceptibility in limbal epithelial cells. **Fig 10** summarizes the effects of PAX6 knockdown and travoprost on the expression of the analyzed genes and proteins in limbal epithelial cells.

Our study has several limitations. The siRNA-mediated knockdown model only partially replicates PAX6 haploinsufficiency and does not account for the genetic heterogeneity observed in patients with congenital aniridia. Additionally, the 20-minute short-term exposure to travoprost may not accurately reflect the long-term effects of chronic medication use in clinical settings. In the present study, the limited amount of available material prevented us from measuring the active, phosphorylated forms of ERK and NF-κB, highlighting the need for further investigation. For the same reason, we were also unable to analyze the Akt and Wnt pathways, which we propose to investigate in future studies. To address these limitations, future studies involving *in vivo* animal models are warranted to further validate and extend our findings.

## 5 Conclusions

In this study, a PAX6-knockdown LEC model was established to systematically evaluate the effects of travoprost on cell viability, proliferation, and migration, as well as the expression of FOSL2, MAPKs, inflammation-related markers, caspase-3, and MMP9 in both normal and PAX6-deficient LECs. The findings revealed that travoprost at specific concentrations inhibits primary human LEC viability and proliferation. Moreover, travoprost may enhance the migratory capacity of PAX6-knockdown LECs through activation of the JNK signaling pathway and upregulation of MMP9. Additionally, it may promote apoptosis via the caspase-3 pathway. Further studies are required to fully elucidate the roles of these signaling mechanisms.

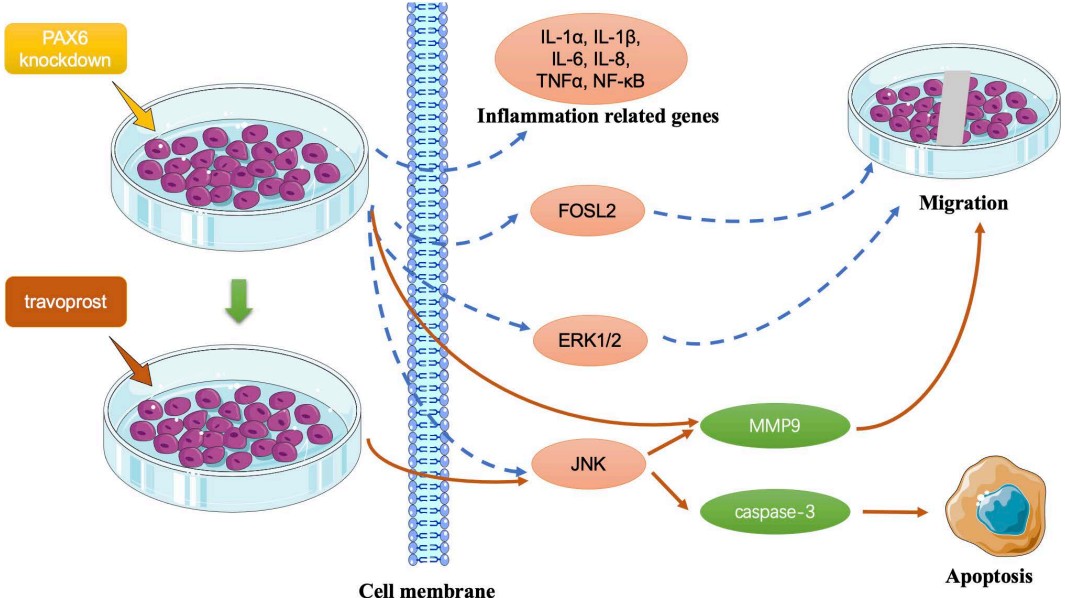

**Fig 10. Effects of PAX6 knockdown and travoprost on the analyzed genes and proteins in limbal epithelial cells (LECs).** PAX6 knockdown inhibits the expression of inflammation-related genes—IL-1α, IL-1β, IL-6, IL-8, TNFα, NF-κB—as well as FOSL2, ERK1/2, and JNK in LECs, while it promotes the expression of MMP9. Travoprost treatment increases the expression of MMP9 and caspase-3 by upregulating JNK levels, thereby enhancing both cell migration and apoptosis. The red solid line indicates an upregulating effect, while the blue dashed line indicates a downregulating effect.

## Acknowledgments

The work of Shuailin Li, Tanja Stachon, Shanhe Liu, Zhen Li, Shao-Lun Hsu, Swarnali Kundu, Fabian N. Fries, Maryam Amini, Shweta Suiwal and Nóra Szentmáry at the Rolf M. Schwiete Center for Limbal Stem Cell and Aniridia Research was supported by the Rolf M. Schwiete Foundation. The work of Shuailin Li, Zhen Li and Shanhe Liu has been supported by the China Scholarship Council. We would like to thank Mrs. Sabrina Häcker for her excellent technical assistance.

## Author contributions

**Conceptualization:** Shuailin Li, Nóra Szentmáry.

**Data curation:** Shuailin Li, Shanhe Liu.

**Formal analysis:** Shuailin Li.

**Investigation:** Shuailin Li, Shanhe Liu.

**Methodology:** Shuailin Li, Tanja Stachon, Nóra Szentmáry.

**Project administration:** Shuailin Li, Tanja Stachon, Nóra Szentmáry.

**Resources:** Tanja Stachon, Nóra Szentmáry.

**Supervision:** Tanja Stachon, Nóra Szentmáry.

**Writing – original draft:** Shuailin Li.

**Writing – review & editing:** Tanja Stachon, Shanhe Liu, Zhen Li, Shao-Lun Hsu, Swarnali Kundu, Fabian N. Fries, Berthold Seitz, Maryam Amini, Shweta Suiwal, Nóra Szentmáry.

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
