## [Decision Letter · Decision Letter 0]

25 Jun 2025

PONE-D-25-21972The Effect of Travoprost on Primary Human Limbal Epithelial Cells and the siRNA-based Aniridia Limbal Epithelial Cell Model, In VitroPLOS ONE

Dear Dr. Li,

Thank you for submitting your manuscript to PLOS ONE. After careful consideration, we feel that it has merit but does not fully meet PLOS ONE’s publication criteria as it currently stands. Therefore, we invite you to submit a revised version of the manuscript that addresses the points raised during the review process.

The manuscript by Li et al. has been reviewed and significant concerns voiced. Please address all issues and pay particular attention to the following points:

1. Please justify the 20 min timepoint selected for travoprost treatment. This seems to be a very short exposure of the drug and does not accurately mimic clinical settings.

2. The authors should discuss why cell viability is decreased even at lower concentrations.

3. The results section needs to be re-written for clarity. The performed experiments need a rationale.

4. Figure 3 images have low resolution. Please increase the resolution when making the tiff file.

5. Please clarify whether travoprost affects cell proliferation in siPAX6 LEC, not just cell migration.

6. Please revisit the Figure 4, 0.313 ug/mL travoprost treatment: resolve the discrepancy between an increase in PAX6 expression and the bar graph.

7. Please provide rationale for measuring the expression of several genes/proteins like FOSL1, ERK1, ERK2, JNK, and MMP-9 that have been shown without any prior mention.

8. In Figures 5 and 6, please comment on why the expression of several markers are changed at the mRNA level but not at the protein level. This could be because the authors have measured the expression of total proteins (e.g., ERK and NF-kB) but not the phosphorylated (activated) forms.

9. There are several pathways that mediate cell proliferation and migration (e.g., Akt or Wnt). The authors have not explained why ERK was selected over other pathways. Please check some other effectors.

10. The Quantitative PCR should be Quantitative RT-PCR.

We look forward to receiving your revised manuscript.

Kind regards,

Alexander V Ljubimov, Ph.D.

Academic Editor

PLOS ONE

Journal Requirements:

Reviewers' comments:

Reviewer's Responses to Questions

**Comments to the Author**

1. Is the manuscript technically sound, and do the data support the conclusions?

Reviewer #1: Partly

2. Has the statistical analysis been performed appropriately and rigorously? 

Reviewer #1: Yes

3. Have the authors made all data underlying the findings in their manuscript fully available?

Reviewer #1: Yes

4. Is the manuscript presented in an intelligible fashion and written in standard English?

Reviewer #1: No

5. Review Comments to the Author

Reviewer #1: The manuscript by Li et al. investigates the in vitro effect of travoprost on primary limbal epithelial cells in an siRNA based aniridia model. The authors have studied an important topic that may be relevant clinically for aniridia treatment. However, there are several concerns that makes it unsuitable for publication in its current form:

1. The authors have not explained how the 20 min timepoint was selected for travoprost treatment. This is a very short exposure of the drug and does not accurately mimic clinical settings.

2. The authors do not discuss why cell viability is decreased even at lower concentrations.

3. The results section needs to be re-written for clarity and flow. The authors have merely stated what experiments are performed without providing a rationale.

4. Figure 3 images are in low resolution. Please provide them in high resolution.

5. It would be beneficial to see if travoprost affects cell proliferation in siPAX6 LEC, and not just on cell migration.

6. In Figure 4, 0.313 ug/mL travoprost treatment seems to increase PAX6 expression, which is not accurately represented by the bar graph.

7. The expression of several genes/proteins like FOSL1, ERK1, ERK2, and JNK have been shown in results without any prior mention. The authors should provide appropriate rationale before the result or mention them within the context of this study in the introduction. Same for MMP9.

8. In Figures 5 and 6, the authors have not explained why the expression of several markers are changed at the mRNA level but not at the protein level. This could be because the authors have measured the expression of total proteins (for eg. ERK and NFkB) and not the phosphorylated (activated) form.

9. There are several pathways that mediate cell proliferation and migration (for eg. Akt or Wnt). The authors have not explained why ERK was selected over other pathways.

6. PLOS authors have the option to publish the peer review history of their article (what does this mean? ). If published, this will include your full peer review and any attached files.

**Do you want your identity to be public for this peer review?** For information about this choice, including consent withdrawal, please see our Privacy Policy .

Reviewer #1: **Yes: ** Ruchi Shah

---

## [Author Response · Author response to Decision Letter 1]

19 Jul 2025

Reviewer #1:

1. Please justify the 20 min timepoint selected for travoprost treatment. This seems to be a very short exposure of the drug and does not accurately mimic clinical settings.

We thank the reviewer for this insightful comment. In vivo studies in rabbits have shown that plasma concentrations of travoprost peak within 10 to 30 minutes after administration and then decline rapidly (Waugh J, et al. Drugs Aging. 2002;19(6):465–473). In our preliminary experiments, we treated limbal epithelial cells (LECs) with various concentrations of travoprost for 10, 20, and 30 minutes, respectively. These tests revealed that a 10-minute exposure had no discernible effect on LEC viability, whereas a 30-minute exposure resulted in a marked reduction in cell viability, which in turn made it challenging to collect sufficient cells for subsequent Western blot and qPCR analyses.

Taking these observations into account, we selected a 20-minute exposure as a representative short-term in vitro treatment model. This duration aligns with several previous in vitro studies that investigated the effects of prostaglandin analogs on corneal epithelial cells (Paimela T, et al. Mol Vis. 2012;18:1189–1196; Whitson JT, et al. Adv Ther. 2012;29(10):874–888). Additionally, one of our prior studies employed a similar 20-minute travoprost treatment to examine effects on limbal stromal cells (Li S, et al. PLoS One. 2025;20(6):e0326967).

Nonetheless, we fully agree that studies involving chronic exposure are essential to more accurately reflect clinical conditions. We have now acknowledged this limitation in the Discussion section and highlighted the need for future in vivo and long-term exposure studies.

2. The authors should discuss why cell viability is decreased even at lower concentrations.

We appreciate the reviewer’s suggestion. We have incorporated the required information into the Discussion section, as follows:

“At lower concentrations, the reduction in LEC viability induced by travoprost may not result from overt cytotoxicity but rather from the early activation of stress- and apoptosis-related signaling pathways. In PAX6-knockdown LECs, we observed an increase in caspase-3 protein levels following treatment with 0.313 μg/mL travoprost, supporting the hypothesis that travoprost can trigger apoptotic processes even at relatively low concentrations. This early response appears to be more pronounced in PAX6-deficient cells, which exhibit impaired regulatory capacity and heightened sensitivity to external stimuli. Similarly, we found that low-concentration travoprost treatment elevated JNK protein levels in PAX6-knockdown LECs, suggesting that travoprost may also induce mild oxidative stress under these conditions [37].”

3. The results section needs to be re-written for clarity. The performed experiments need a rationale.

Thank you for this valuable feedback. We have restructured the Results section to enhance clarity by introducing a clear rationale at the beginning of each subsection. These revisions are intended to guide the reader more effectively through the experimental design and the interpretation of our findings.

4. Figure 3 images have low resolution. Please increase the resolution when making the tiff file.

Thank you very much. We have replaced Figure 3 with higher-resolution TIFF images to meet publication standards, and these updated images are included in the revised submission.

5. Please clarify whether travoprost affects cell proliferation in siPAX6 LEC, not just cell migration.

Thank you for pointing this out. Performing PAX6 knockdown in 96-well plates presents certain technical challenges, which is why we did not assess the effect of travoprost on the proliferative capacity of PAX6-knockdown LECs in this study.

However, we have now included a comparative analysis of Ki67 mRNA levels following travoprost treatment in the revised Results section (Figure 5). While the primary suppression of proliferation was observed in normal LECs at higher travoprost concentrations, no statistically significant changes in Ki67 mRNA levels were detected in either LECs or PAX6-knockdown LECs treated with 0.156 or 0.313 μg/mL travoprost. This clarification has been incorporated into both the Results and Discussion sections.

“3.4 Ki67 mRNA levels

Ki67 is widely regarded as a marker of cell proliferation [32]. The mRNA levels of Ki67 did not differ significantly between LECs and PAX6-knockdown LECs (p = 0.321). Moreover, treatment with travoprost did not affect Ki67 mRNA expression in either the LEC or PAX6-knockdown subgroups (p ≥ 0.867) (Fig 5).

Fig 5. Ki67 mRNA levels in limbal epithelial cells (LECs) and PAX6-knockdown LECs following a 20-minute treatment with travoprost at concentrations of 0.156 μg/mL and 0.313 μg/mL (n=7). mRNA values are presented on a logarithmic scale (Log₂) and expressed as the geometric mean±geometric standard deviation (SD). Two-way ANOVA was used for statistical analysis, and significant p-values (p<0.05) are indicated in the diagrams.

No significant differences were observed between any of the analyzed groups or subgroups (p≥0.321).”

“We also examined the effects of 0.156 and 0.313 μg/mL travoprost on Ki67 mRNA levels, a marker of cell proliferation [32], in both LECs and PAX6-knockdown LECs. These travoprost concentrations did not affect proliferation in either group, consistent with our BrdU assay results.”

6. Please revisit the Figure 4, 0.313 ug/mL travoprost treatment: resolve the discrepancy between an increase in PAX6 expression and the bar graph.

We thank the reviewer for bringing this to our attention. We found no significant increase in PAX6 expression after treatment with 0.313 μg/mL travoprost in either LECs or PAX6-knockdown LECs. Figure 4 now includes a representative Western blot image for clarity.

7. Please provide rationale for measuring the expression of several genes/proteins like FOSL1, ERK1, ERK2, JNK, and MMP-9 that have been shown without any prior mention.

Thank you very much. We appreciate this suggestion and have now revised the Introduction to more clearly justify our focus on these targets:

“Fos like 2 (FOSL2) is considered a target gene of PAX6 and is closely associated with corneal opacity [21]. It is also thought to play a role in regulating cell migration and proliferation [22]. Mitogen-activated protein kinases (MAPKs) are primarily categorized into three subfamilies: extracellular signal-regulated kinases (ERKs), c-Jun N-terminal kinases (JNKs), and mitogen-activated protein kinase 14 (MAPK14, also known as p38) [23]. Among these, ERKs and JNKs are particularly implicated in controlling cell proliferation, migration, and apoptosis [24–26]. As previously noted, travoprost can elevate MMP levels, thereby contributing to the reduction of IOP [27]. Related studies have also demonstrated that MMP expression is regulated through the JNK pathway [28]. In our earlier investigation of aniridia limbal stromal cells (AN-LSCs), we found that travoprost may influence MMP9 expression via the JNK signaling pathway, subsequently affecting cell migration [29]. Based on these findings, the present study focuses on evaluating the expression of these key genes and proteins.”

8. In Figures 5 and 6, please comment on why the expression of several markers are changed at the mRNA level but not at the protein level. This could be because the authors have measured the expression of total proteins (e.g., ERK and NF-kB) but not the phosphorylated (activated) forms.

Thank you for this important observation. The reviewer correctly noted the discrepancies where certain markers showed differences at the mRNA level but not at the protein level. We acknowledge that for some proteins, such as ERK and NF-κB, we only measured total protein levels and did not assess their phosphorylated (active) forms. Given that we observed no differences between the subgroups treated with varying concentrations of travoprost, we did not extend our analyses in the present study to include phosphorylated protein levels. Nonetheless, we have now added a note in the Discussion section acknowledging this as a limitation of our study, emphasizing that the active (phosphorylated) forms of ERK and NF-κB were not assessed and warrant further investigation:

“In the present study, the limited amount of available material prevented us from measuring the active, phosphorylated forms of ERK and NF-κB, highlighting the need for further investigation.”

9. There are several pathways that mediate cell proliferation and migration (e.g., Akt or Wnt). The authors have not explained why ERK was selected over other pathways. Please check some other effectors.

We thank the reviewer for this valuable suggestion. Our initial focus on ERK and JNK was based on their well-established roles in ocular epithelial wound healing and their reported regulation by prostaglandin analogs. We agree that examining additional pathways would provide a more comprehensive understanding. Nevertheless, working with primary cells provided only limited material, which made it unfeasible to analyze all pathways in this study. Accordingly, we have added a note in the Discussion acknowledging as a limitation of our current study that neither the Akt nor Wnt pathways were analyzed, and we propose investigating these pathways in future research:

“For the same reason, we were also unable to analyze the Akt and Wnt pathways, which we propose to investigate in future studies.”

10. The Quantitative PCR should be Quantitative RT-PCR.

Thank you for pointing this out. We have corrected all instances of "Quantitative PCR" to "Quantitative RT-PCR" throughout the manuscript.

---

## [Decision Letter · Decision Letter 1]

4 Aug 2025

The Effect of Travoprost on Primary Human Limbal Epithelial Cells and the siRNA-based Aniridia Limbal Epithelial Cell Model, In Vitro

PONE-D-25-21972R1

Dear Dr. Li,

We’re pleased to inform you that your manuscript has been judged scientifically suitable for publication and will be formally accepted for publication once it meets all outstanding technical requirements.

Kind regards,

Alexander V Ljubimov, Ph.D.

Academic Editor

PLOS ONE

Additional Editor Comments (optional):

No further concerns.

Reviewers' comments:

Reviewer's Responses to Questions

**Comments to the Author**

1. If the authors have adequately addressed your comments raised in a previous round of review and you feel that this manuscript is now acceptable for publication, you may indicate that here to bypass the “Comments to the Author” section, enter your conflict of interest statement in the “Confidential to Editor” section, and submit your "Accept" recommendation.

Reviewer #1: All comments have been addressed

2. Is the manuscript technically sound, and do the data support the conclusions?

Reviewer #1: Yes

3. Has the statistical analysis been performed appropriately and rigorously? 

Reviewer #1: Yes

4. Have the authors made all data underlying the findings in their manuscript fully available?

Reviewer #1: Yes

5. Is the manuscript presented in an intelligible fashion and written in standard English?

Reviewer #1: Yes

6. Review Comments to the Author

Reviewer #1: The authors have adequately addressed all concerns. The revised manuscript reads much more clearly than before. No further comments.

7. PLOS authors have the option to publish the peer review history of their article (what does this mean? ). If published, this will include your full peer review and any attached files.

**Do you want your identity to be public for this peer review?** For information about this choice, including consent withdrawal, please see our Privacy Policy .

Reviewer #1: **Yes: ** Ruchi Shah

---

## [Editor Report · Acceptance letter]

PONE-D-25-21972R1

PLOS ONE

Dear Dr. Li,

I'm pleased to inform you that your manuscript has been deemed suitable for publication in PLOS ONE. Congratulations! Your manuscript is now being handed over to our production team.

Kind regards,

on behalf of

Dr. Alexander V Ljubimov

Academic Editor

PLOS ONE